# Carbonyl Cyanide 3-Chloro Phenyl Hydrazone (CCCP) Restores the Colistin Sensitivity in *Brucella intermedia*

**DOI:** 10.3390/ijms24032106

**Published:** 2023-01-20

**Authors:** Malak Zoaiter, Zaher Zeaiter, Oleg Mediannikov, Cheikh Sokhna, Pierre-Edouard Fournier

**Affiliations:** 1Institut Hospitalo-Universitaire Méditerranée-Infection, 13005 Marseille, France; 2Institut de Recherche pour le développement (IRD), Assistance publique des hôpitaux de Marseille (AP-HM), SSA, Vecteurs Infections Tropicales et Méditerranéennes (VITROME), Aix-Marseille Université, 13005 Marseille, France; 3Department of Biology, Faculty of Sciences, Lebanese University LU, Beirut 146404, Lebanon; 4Institut de Recherche pour le développement (IRD), Assistance publique des hôpitaux de Marseille (AP-HM), Microbes, Evolution, Phylogénie et Infection (MEPHI), Aix-Marseille Université, 13005 Marseille, France; 5Campus Commun UCAD-IRD of Hann, Dakar 1020, Senegal

**Keywords:** intrinsic colistin resistance, efflux pump inhibitors, carbonyl cyanide 3-chloro phenyl hydrazone, CO-MIC, synergy, efflux pump, biofilm formation

## Abstract

*Brucella intermedia* (formerly *Ochrobactrum intermedium*), a non-fermentative bacterium, has been isolated from animals and human clinical specimens. It is naturally resistant to polymyxins, including colistin (CO), and may cause opportunistic infections in humans. We isolated six *Brucella intermedia* strains from Senegalese monkey stool. In order to determine whether an efflux pump mechanism was involved in CO resistance in *B. intermedia,* we evaluated the effects of verapamil (VRP), reserpine (RSP), phe-arg β-naphthylamide dihydrochloride (PAβN) and carbonyl cyanide 3-chloro phenyl hydrazone (CCCP), four efflux pump inhibitors, on these colistin-resistant strains. Using the broth microdilution method, a CO and CCCP combination of 2 µg/mL and 10 µg/mL, respectively, significantly reduced the CO minimal inhibitory concentration (MIC) of *B. intermedia*, supporting an efflux pump mechanism. In contrast, VRP, PAβN and RSP did not restore CO susceptibility. A time kill assay showed a bactericidal effect of the CO–CCCP combination. Genomic analysis revealed a potential implication in the CO resistance mechanism of some conserved efflux pumps, such as YejABEF, NorM and EmrAB, as previously reported in other bacteria. An inhibitory effect of the CO–CCCP combination was observed on biofilm formation using the crystal violet method. These results suggest that the intrinsic CO resistance in *Brucella intermedia* is linked to an efflux pump mechanism and that the synergistic effect of CO–CCCP may open a new field to identify new treatments to restore antibiotic efficacy in humans.

## 1. Introduction

Colistin (CO) was isolated in 1947 from the Gram-positive bacterium *Bacillus polymyxa* [1]. Since 2005, it has been recognized as being bactericidal against Gram-negative aerobic bacteria and several *Mycobacterium* species [2]. Consequently, CO has become a key antibiotic, alone or in combination with other antimicrobial agents [3], to treat multidrug resistant (MDR), especially carbapenemase-producing bacteria [3]. CO binds to the negatively charged lipopolysaccharides (LPS) of Gram-negative bacteria (GNB) and penetrates the outer membrane by a “self-promoted uptake” mechanism. This leads to a destabilization of the inner membrane integrity, eventually causing cell death [4]. Resistance to CO may be explained by several mechanisms: (i) The most common is the modification of LPS, the main target of CO. LPS is able to reduce the negative charge of the bacterial surface, and thus the electrostatic interactions with CO [5]. Such an alteration targets the chromosomally encoded genes associated with LPS modification by activating mutational [6] or non-mutational (adaptive) effects in response to environmental signals such as the presence of CO or other inducers [7]. (ii) CO-degrading enzymes have been identified in CO-producing *Bacillus polymyxa* for self-protection [8] and recently in *Bacillus lichenformis* [9], but they have never been linked to CO resistance in other micro-organisms [10]. (iii) Since 2015, plasmid-borne mobilized CO resistance (mcr) genes have been reported, initially in China [11]. mcr genes play a role in bacterial outer membrane remodeling and CO resistance. Such plasmids are easily transmitted and increase the spread of CO resistance. (iv) Other strategies include capsule polysaccharide shielding, as in *Klebsiella pneumoniae* [12], and the efflux pump system related to proton motive force, as was reported in *Acinetobacter* sp. and *Pseudomonas* sp. [13,14].

*Brucella intermedia* (formerly *Ochrobactrum intermedium*) is a non-fermentative bacterium that is widely distributed in soil, plants, water, animals and polluted or industrial environments [15]. Its presence in hospitals and its multi-resistance to antibiotics led to its increasingly frequent isolation in human infections, in particular in immunocompromised patients [16] and cystic fibrosis [17]. It is associated with a variety of infections including bacteremia [18], pelvic abscess [19], dyspepsia [20], endocarditis [21] and pneumonia [17]. *B. intermedia* is naturally resistant to polymyxins, including CO [22]. The resistance mechanism in *B. intermedia* is thus probably mediated by the cell envelope [23]. The export of antimicrobial agents through the activity of efflux pumps plays an important role in both the intrinsic and acquired resistance in MDR bacteria. Recently, several studies have demonstrated that the resistance to CO was reversed by carbonyl cyanide 3-chloro phenyl hydrazone (CCCP) in *Enterobacteriaceae* [23], *Acinetobacter baumannii* [24] and *Stenotrophomonas maltophilia* [25]. CCCP is a strong uncoupler that disrupts proton motive force by affecting the electrochemical transmembrane gradients and reducing the ATP production. Its use was restricted to experimentally testing the active efflux or the biofilm formation [25,26] in resistant bacteria. However, the mechanism of CO–CCCP synergy is still unknown. To date, one report has shown the direct antibacterial activity of CCCP against *Mycobacterium abscessus* [27].

In this article, we investigated the role of efflux pumps in the intrinsic mechanism of CO resistance in *B. intermedia*, isolated from healthy chimpanzee stool, by using CCCP, verapamil (VRP), reserpine (RSP) and phe-arg β-naphtylamide dihydrochloride (PAβN) as efflux pump inhibitors (EPI).

## 2. Results

### 2.1. Bacterial Isolation and Identification

Six *B. intermedia* strains (Q1103, Q1105, Q1107, Q1108, Q1109 and Q1111) and one *B. tritici* strain (Q1106) were cultivated from the stool sample and identified by MALDI-TOF MS. A total of 16S rRNA and *recA* genes confirmed the affiliation of the five strains to *B. intermedia*. A total of 16S rRNA sequences showed a similarity of >98.8% with those of *B. intermedia* strains available in NCBI. *rec*A gene sequences exhibited a >98.5% similarity with *B. intermedia.* Phylogenetic trees based on the 16S rRNA and *recA* genes confirmed that all isolated *B. intermedia* were clustered together (Figure 1).

Strain Q1103 was phenotypically different from other isolated *Brucella* strains by an important production of an extracellular matrix around colonies. The Q1105, Q1107, Q1108, Q1109 and Q1111 strains were phenotypically similar. Thus, we selected two strains that were phenotypically different, Q1103 and Q1105 (chosen randomly among the five remaining strains), for the following analyses.

### 2.2. CO, CCCP and CO/CCCP MIC Determination

Using the broth microdilution, all six *B. intermedia* isolates showed a CO-MIC of >512 µg/mL, confirming their intrinsic resistance (Figure 2a and Appendix A). VRP, RSP and PaβN did not cause any decrease in CO MIC of *B. intermedia* at the tested concentrations (Appendix A). Therefore, only results for CCCP were discussed. For the protonophore CCCP, we obtained distinct MICs, even for a given strain, by repeating the experiment 10 times. Only the most frequent MIC for each strain was noted in Table 1. This led us to think that the population within a *Brucella* strain may be heterogeneous with regard to response to CCCP. A successful isolation of colonies for each strain that respond to different concentrations of CCCP (5, 6, 7, 8, 9, 10 and 15 µg/mL) confirmed our hypothesis. As a result, we divided each strain into two populations with regard to the used concentration in the literature (10 µg/mL of CCCP) [28]. The first including colonies that were able to survive at 10 µg/mL CCCP and the second were those exhibiting a CCCP MIC < 10 µg/mL. In our study, we focused on the colonies with higher CCCP MIC (>10 µg/mL). Most of the cells within strain Q1103 exhibited a CCCP MIC of <30 µg/mL; for strains Q1105, Q1107 and Q1109, the majority of MICs were <20 µg/mL; for strains Q1111 and Q1108, MICs were most commonly <15 µg/mL (Table 1). We then assessed the effect of the efflux pump on the antibacterial susceptibility profile to CO by evaluating the CO MIC in the presence of CCCP. The addition of the CCCP inhibitor significantly reduced the MIC of CO to <2 µg/mL, with a mean fold-change of 426.66, which is more than an 8-fold change, confirming the involvement of the efflux pump in the resistance mechanism (Table 1). The FICI of the combination was lower than 0.5, confirming the synergism of the two drugs with the chosen concentration of CCCP (10 µg/mL). Table 1 summarizes the CO MICs alone as well as with CCCP, the MIC fold change and the fractional inhibitory concentration. The effect of CCCP on the CO MIC was also assessed on agar plates. All plates showed that CCCP had an effect alone on *Brucella* growth. Likewise, CCCP revived the activity of CO by the appearance of the circular inhibition zone. The diameters were measured at between 21 and 23 mm for all strains, except Q1103, which exhibited a diameter of 19 mm (Figure 2b and Appendix A). The *E. coli* strain, used as a positive control, was resistant to CO and has a known CO MIC of 4–8 µg/mL, determined by broth microdilution [28]. The strain become susceptible to CO with an MIC of <0.5 µg/mL when 10 µg/mL of CCCP was added. On agar plates, the diameter of the CO inhibition zone increased from 12 mm to 15 mm on plates containing CCCP. CCCP alone had no effect on the used *E. coli* strain, either in broth or in agar (Figure 2a,b).

### 2.3. Synergistic Effect of CCCP-CO Inhibits Bacterial Growth

To confirm the obtained MIC results and to choose the best CCCP–CO combination, we monitored the growth kinetics of the treated bacterial strain cultures in a microtiter plate. The control curves of strains Q1105 and Q1103 showed a prolific growth as the populations reached the maximum OD of 1.5 after 24 h (Figure 3). Increasing concentrations of CO did not affect significantly the growth of both strains (*p* < 0.0388; *p* = 0.999) (Figure 3a,b). Furthermore, the addition of CCCP at different concentrations resulted in variable levels of growth inhibition of both strains. CCCP treatment showed a remarkable dose-dependent effect, with an increase in lag phase duration and a decrease in the maximum specific growth rate (*p* < 0.0001) (Figure 4c,e). Complete growth inhibition was observed at 20 µg/mL and 30 µg/mL in strains Q1105 and Q1103, respectively, confirming the MIC results. Growth was suppressed from 2 µg/mL of CO with CCCP for both Q1103 and Q1105 strains (*p* < 0.0001) (Figure 4d,f). Unlike CCCP, VRP, RSP and PAβN exhibit an incomplete inhibitory effect in combination with CO on *B. intermedia* growth (Appendix A). These growth inhibition assays not only confirmed our MIC results, but also showed that the synergistic effect of CO/CCCP could be bactericidal too. Based on our findings, we decided to perform a time-kill assay with the combination of 2 µg/mL CO and 10 µg/mL CCCP.

### 2.4. Bactericidal Effect of CCCP–CO

Due to the heterogeneous treatment responses to CCCP, time-kill curves were assessed for strains Q1105 and Q1103 which represent both populations (CCCP MIC > 10 µg/mL and CCCP MIC < 10 µg/mL, respectively). For population 1, exhibiting a CCCP MIC of >10 µg/mL, both Q1105 and Q1103 *Brucella* strains had lower growth rates than the controls when treated with CCCP alone (Figure 5a,c). However, after 24 h, a weak bactericidal effect, estimated to be 19.2% for Q1103 and 29.2% for strain Q1105, was observed in the presence of the CO/CCCP combination. This effect increased much more in strain Q1105 after 48 h (23.5% for strain Q1103 and 62.8% for strain Q1105). For population 2 (CCCP MIC < 10 µg/mL), a bactericidal effect was shown for CCCP alone in strains Q1103 and Q1105 (67.8% and 74.5%, respectively) after 24 h, which increased after 48 h (91% and 99%, respectively) (Figure 5b,d). A complete bactericidal effect was obtained with the CO/CCCP combination after 24 h in both strains.

### 2.5. CO-Treatment with CO/CCCP Inhibits Biofilm Formation of B. intermedia

To explain the difference in bactericidal effect between the two strains after 48 h in population 1, we considered the phenotypic differences between them. Strain Q1103 was able to form a matrix on agar plates much more efficiently than strain Q1105, which can be considered a biofilm layer. Therefore, biofilm formation assays were carried out using the crystal violet (CV) method. After 48 h of incubation time, we found that strain Q1103 adhered to the culture plate much more than strain Q1105 (*p* < 0.0003). Biofilm formation capability of both strains was not affected when treated with 10 µg/mL of CCCP or with increasing concentrations of CO (Figure 6). Furthermore, treatment of both strain cultures with various CO/CCCP combinations yielded a dose-dependent decrease in their biofilm formation ability; where a clear and significant decrease began at the CO/CCCP combination of 10 µg/mL and 2 µg/mL (*p* < 0.0001). The inhibitory effect between the CCCP-treated and the non-CCCP-treated groups was statistically significant for strains Q1103 and Q1105 (*p* < 0.0001 and *p* < 0.0014, respectively). Taken together, our data suggest that *B. intermedia* biofilms may play a crucial role in CO resistance.

### 2.6. Pangenomic Analysis

Based on our results, the addition of CCCP in combination with CO rescued the intrinsic resistance to CO in studied *B. intermedia* strains. This led us to assume that efflux pumps may be involved in their resistance mechanism. Through genomic analysis, we decided to identify the common genes encoding for efflux pumps that can be involved in the CO resistance mechanism. A pangenome analysis showed that *B. intermedia* genomes contained 8276 gene clusters, and 3546 of these comprised the core genome in which conserved putative efflux pumps were found (Figure 7). Some of these have been experimentally shown to be involved in the resistance to polymyxins, including CO, in other bacteria. Among the core genome cluster of *B. intermedia*, we found ATP-Binding (ABC) transporters such as YejABEF and LolCDE. These effluxes were reported in *Brucella melitensis* and *Acinetobacter baumannii,* respectively [29,30]. NorM, which is a member of multidrug and toxic efflux (MATE) family, was detected in the core genome. It contributes to the polymyxin B intrinsic resistance in *Burkholderia Vietnamiensis* [31]. Furthermore, EmrE, a small multidrug resistance (SMR) efflux pump, was conserved within the studied strains and showed more than 60% similarity with that of Burkholderia stabilis, representing a multi-resistant transporter of cationic drugs (www.uniprot.org). Another Emr pump, EmrAB, that is known to form a tripartite complex with the outer membrane protein TolC, was also found in *Brucella* strains. It belongs to the major facilitator superfamily (MFS) and was reportedly involved in a CO resistance mechanism in *Acinetobacter baumannii* [13]. *Brucella* genomes contained, in addition, a conserved permease ATP-dependent MapB, a TamB ortholog that has been demonstrated to play a role in CO resistance in *Brucella suis* [32]. Other conserved multiresistant efflux pumps that may be involved in the CO resistance mechanism include BepE, YbhRFS, Mdtk, Bmr3, EmrK and OqxB28.

## 3. Discussion

In recent years, CO has been receiving much attention in response to an increasing prevalence of antibiotic resistance worldwide [33]. In human medicine, CO has been used as last resort drug to treat MDR bacteria, especially carbapenem-resistant bacteria. CO is inactive against intrinsically resistant bacteria, including *Brucella* species, which suggests a role of the cell wall in this mechanism [34]. The conventional mechanisms of intrinsic resistance include the outer membrane impermeability to antibiotics and the activity of multidrug efflux pumps that prevent the access of the drugs to their targets [35]. Several EPI have been evaluated to re-establish CO activity, such as VRP, RSP, PABN and CCCP. In our study, VRP, PAβN and RSP failed to restore the CO activity in *B. intermedia,* confirming previous studies on MDR-GNB [25]. In contrast, a potential synergy between CCCP and CO was observed in reversing CO resistance in *B. intermedia,* as it was reported in *A. baumannii*, *Stenotrophomonas maltophilia* [25] and *Enterobacteriaceae* [34]. CCCP reduced CO MIC more than 256-fold, suggesting that the mechanism inhibited by CCCP is essential for their CO resistance. CCCP, a potential nonspecific protonophore uncoupler, acts by dissipating PMF, the most common source of energy in bacterial cells, and subsequently inhibiting their efflux pump activity [36]. Our findings strongly suggest the efflux pump inhibitory effect of CCCP. Time-kill curves showed that CCCP had a bactericidal effect alone and in combination with CO. This can be explained by the fact that CCCP and CO may share common efflux pumps that are essential for *Brucella* growth.

It should be noted that CCCP alone showed a heterogenous inhibitory effect in tested *Brucella* strains. Such a heterogeneity in CCCP response was reported recently in *E. coli* and explained by a heterogenous dissipation of PMF between cells [37]. The heterogeneity in response to CCCP that was observed among cells of a given strain may be explained by the fact that the molecule targets efflux pumps that act on itself and exclude it from cells with high efflux pump activity. Therefore, these cells maintain their PMF and the efflux pump activity, generating a positive feedback between CCCP and efflux pumps [37].

Based on the pangenome analysis, several efflux pumps, such as YejABEF, LolCDE and NorM, were identified in the *Brucella* core genome. Previous studies have found their involvement in CO resistance in other organisms [13,29]. *Emr* (MFS), especially EmrA, a part of the tripartite EmrAB-TolC efflux pump, was demonstrated to confer resistance to CCCP in *E. coli* and to CO in *A. baumannii.* This may explain the common bactericidal effect of CCCP alone and combined with CO [13,38]. TamB provided resistance to CO in *Brucella suis*. It should be noted that the Mex and Acr efflux pumps that contributed to CO resistance in *Pseudomonas aeruginosa* and *Acinetobacter baumannii* were detected in some *Brucella* strains, but as they were not located in the core genome, their involvement in CO resistance is unlikely [30,39].

As CCCP blocks PMF and ATP synthesis, it affects, in addition to the efflux pump activity, the cellular metabolism in all energy requiring reactions [36]. Subsequently, we detected several conserved enzymes that confer resistance to polymyxins, such as those involved in the biosynthesis of lipopolysaccharides WadA, WadC [40], LpxM or MsbB [41] and EptA [42] and in carbohydrate pathways such as pgm [43].

Biofilm formation can be a second factor in the CO resistance mechanism. The inhibitory effect of the CO–CCCP combination observed using the crystal violet method may either be explained by an inhibition of the efflux pumps involved in the biofilm formation or by the bactericidal effect induced by the CCCP–CO combination. To elucidate the potential effect of CCCP on biofilm formation and its relation to CO resistance, more experimental analyses, including analysis of the biomass and the extracellular matrix of treated cells, should be carried out.

In summary, we demonstrated for the first time that CO resistance in *B. intermedia* can be reversed by CCCP. The precise mechanism of this antibiotic efficiency restoration requires further investigation, especially in the elucidation of the specific role of each efflux pump system identified by the pangenome analysis. Additional work is also needed to clarify the effect of CCCP on biofilm formation in *B. intermedia.* Due to its intrinsic cytotoxicity, this molecule cannot be applied in the clinical field but may serve as a model to identify new agents that potentiate the activity of colistin for the treatment of CO-resistant infections. 

## 4. Materials and Methods

### 4.1. Bacterial Isolation and Culture Conditions

Six Brucella strains were isolated from chimpanzee stool collected on the ground on 6 June 2016, in the Dindefelo Community Nature Reserve (12°22′01.4″ N 12°18′00.0″ W) in south-eastern Senegal. Stools were externally decontaminated with 70% ethanol prior to culture. Then, they were enriched in Tryptone Soy Broth TSB (bioMérieux, Marcy-l’Étoile, France) at 37 °C for 72 h. Ten microliters of culture were inoculated on CO TSB agar. Four micrograms of CO/mL was chosen based on the European Committee on Antimicrobial Susceptibility Testing (EUCAST) guidelines. The plates were then incubated for 24 h at 37 °C. Individual colonies were replicated on 5% sheep blood-enriched Columbia agar (COS, bioMérieux), and then identified by the matrix assisted laser desorption/ionization time-of-light mass spectrometry MALDI-TOF MS (Bruker Daltonics, Leipzig, Germany). The study of strains was permitted by the Senegalese Nagoya authority under reference 001042. The strains were deposited in the CSUR collection as follows: Q1103, Q1105, Q1107, Q1108, Q1109 and Q1111.

### 4.2. Genome Sequencing, Annotation and Phylogenetic Analyses

Genomic DNA (gDNA) of Brucella strains was extracted using the EZ1 biorobot with the DNA Tissue kit, as recommended by the manufacturer (Qiagen, Hilden, Germany). gDNA was quantified by a Qubit assay with the highly sensitive dsDNA kit (Invitrogen, Carlsbad, CA, USA). Sequencing was performed with the Miseq technology and the mate-pair strategy (Illumina Inc., San Diego, CA, USA), as previously described [44]. Genomes were assembled using Spades version 3.14.0 and were annotated by PROKKA version 1.14.5. Pangenomic analysis was performed using Roary software version 3.10.5 and was visualized on the available Phandango website phandango (jameshadfield.github.io) accessed on March 2021 [44,45,46]. A total of 16S rRNA sequences were extracted from the genomes using contEst16S EZbiocloud ContEst16S|Ezbiocloud.net. Two *B. intermedia* genomes were downloaded from NCBI (NCTC12171 and LMG3301) to be compared to those of chimpanzee strains. Their accession numbers were UGSH00000000.1 and ACQA00000000.1, respectively. *RecA* gene sequences, also extracted from genome sequences, were used to confirm the affiliation of the strains. The sequences were aligned by Clustal W [47]. Evolutionary analyses were inferred using the maximum likelihood method based on the Tamura-Nei model, and were conducted in MEGA version 7.0.26 [48].

### 4.3. Chemicals and Stock Solution

CO sulphate salt (CO) was purchased from MP-Biomedicals (Illkirch-Graffenstaden, France). All efflux pump inhibitors, CCCP, verapamil (VRP), reserpine (RSP) and phe-arg β-naphthylamide dihydrochloride (PAβN) were purchased from Sigma Aldrich (Lyon, France). CCCP and RSP were dissolved in DMSO at a concentration of 5 mg/mL. VRP was dissolved in ethanol 99% in a concentration of 5 mg/mL. PAβN was diluted to 5 mg/mL in deionized water. All stock solutions were preserved at −20 °C. The final concentration of PaβN, RSP and VRP was 50 µg/mL.

### 4.4. Minimal Inhibitory Concentration (MICs) Determination

Currently, the broth microdilution method is the recommended technique to determine the susceptibility to CO [49]. Susceptibility testing was performed according to the EUCAST guidelines using the cation-adjusted Mueller Hinton broth CAMHB (BD-ThermoFisher, Bourgoin-Jallieu, France). MICs were evaluated in the presence of CO alone and with the four efflux pump inhibitors. CO was diluted to final concentrations ranging from 0.25 to 256 µg/mL. Efflux pump inhibitor concentrations ranged from 2.5 to 100 µg/mL. To test the effect of efflux pump inhibitors on the CO MIC of the tested strains, a sub-MIC of 10 µg/mL for CCCP and 50 µg/mL for each of VRP, RSP and PAβN were combined with the various CO concentrations mentioned above [28,50].

A mcr-mediated CO-resistant *E. coli* strain, used as positive control, became susceptible when adding 10 µg/mL CCCP, as previously described [28].

For each tested strain, the inoculum was prepared by resuspending three to five colonies in 0.85% NaCl medium to obtain the standard turbidity using MacFarland standard 0.5. Then, 5 × 10^5^ CFU/mL was added in each well of a 96-well plate containing the tested antibiotic dilutions. Plates were incubated at 37 °C and MIC results were visualized after 16 h–18 h by adding iodonitrotetrazolium (Sigma-Aldrich S.A.R.L, Lyon, France). MIC testing for each strain was carried out in duplicate in three independent experiments.

### 4.5. MIC Fold Change and Synergy Index

The MIC fold change after CO–CCCP combination was defined by the ratio of the CCCP-free CO to that of CCCP-added CO. The mean fold change was calculated following the equation:[1/n] × Σ[MIC fold change x frequency of fold change]

The influence of an efflux pump on the CO-MIC for a bacterial strain was defined as an at least 8-fold reduction in MIC in the presence of CCCP [28]. The effect of CO appeared to be reversed when a strain became susceptible with an MIC of ≤2 µg/mL, according to EUCAST [50,51]. The interaction between the two molecules was determined by calculating the fractional inhibitory concentration index (FICI). The fractional inhibitory concentration, FIC, was derived from the lowest concentration of CO and CCCP combination permitting no visible growth of *Brucella intermedia* in the wells. The FIC value for each agent was calculated according to the formula:FIC CO = (MIC of combination/MIC of CO)
FIC CCCP = (MIC of combination/MIC of CCCP)

To calculate the FIC index, the following formula was used: FICI = FIC (CO) + FIC (CCCP). The combination was classified as synergistic if the FIC indices were lower than or equal to 0.5; additive if the FIC indices were greater than 0.5 and lower than or equal to 1; indifferent if the FIC was greater than 1 and lower than or equal to 2; and antagonistic if the FIC was greater than 2 [52].

### 4.6. Effect of CCCP on CO MIC on Agar Plates

The effect of CCCP on CO MIC was assessed on CAMH agar plates. CCCP was added at a final concentration of 10 µg/mL. Commercial diffusion 50 µg CO disks were deposited on the plates (bioMérieux). Plates without CCCP were prepared with DMSO 0.2%. Agar plates were incubated under aerobic conditions at 37 °C for 18 h to 24 h. The density of the inoculum was standardized at 0.5 MacFarland. All experiments were performed in triplicate.

### 4.7. Crystal Blue Assay

*Brucella intermedia* culture was performed using the same method of MIC microdilution in a 96-well polystyrene microtiter plate. Biofilm formation was assessed under static growth in Mueller Hinton broth. After 48 h of incubation, the supernatant was removed. Wells were washed with phosphate-buffered saline (PBS) three times to remove the planktonic bacteria. The plate was dried well. Adherent biofilms were stained with 125 μL of 0.1% crystal violet. The microtiter plate was incubated for 10–15 min at room temperature. The dye was removed and three washes were performed with water to eliminate all excess cells and dye. The plate was turned upside-down and air-dried for a few hours. Dye that was bound to adherent cells was re-solubilized with 125 μL of 30% acetic acid in water for 10–15 min. The absorbance was measured at 550 nm with a plate reader using 30% acetic acid in water as a blank. Three biological replicates were performed to confirm reproducibility.

### 4.8. Growth Curves

Plates were prepared as described above.

Bacterial growth was measured by optical density (OD 630) using a Gen-5 microplate reader (Agilent-Biotek, Les ULIS, France). CO concentration ranged from 256 to 0.5 µg/mL. CCCP concentration was fixed at 10 µg/mL. PaβN, RSP and VRP were used at 50 µg/mL. Strains Q1103 and Q1105 were grown in Muller Hinton broth at 37 °C with shaking continuously for 24 h to 48 h. CO and efflux pump inhibitors were added at the beginning of the experiment.

### 4.9. Time-Kill Assay

To explore the effect of CCCP alone and in combination with CO, time-kill assays were performed. Two strains were chosen according to their phenotypic characteristics (Q1103 was more viscous than Q1105). An amount of 5 × 10^5^ CFU/mL were inoculated in MHCA broth and were incubated for 24 h and 48 h with shaking. Time-kill analysis was performed by adding a combination of 10 µg/mL CCCP and 2 µg/mL CO.

### 4.10. Statistical Analysis

Data were represented as the means ± standards deviations for three independent experiments and replicated twice. Data were analyzed using the GraphPad Prism version 8 software. Significant differences among the various treatment conditions were evaluated using the one way analysis of variance (ANOVA) and multiple comparisons were performed by the non-parametric student’s t-test. Differences were considered to be significant at *p* < 0.05.

## Figures and Tables

**Figure 1 ijms-24-02106-f001:**
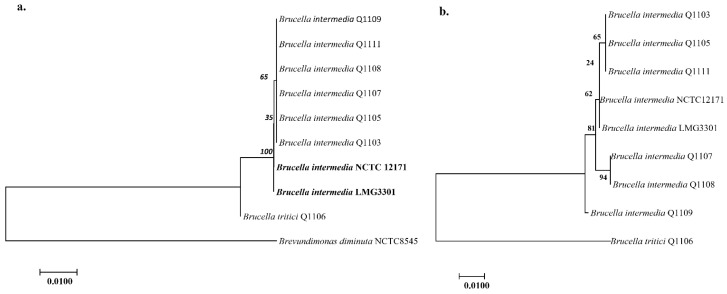
Phylogenetic trees showing the clustering of *B. intermedia* strains based on 16S rRNA (**a**) and RecA (**b**) genes using the maximum likelihood method. The scale bar represents a 1% sequence divergence. *Brevundimonas diminuta* NCTC 8545 was used as outgroup for the 16S rRNA gene. Species in bold letters indicate those for which genomes were downloaded from GenBank.

**Figure 2 ijms-24-02106-f002:**
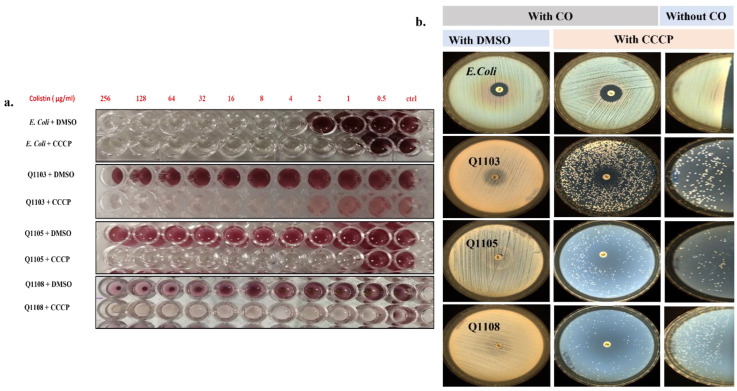
Observation of the effect of CCCP alone and in combination with CO using the liquid method (**a**) and the agar method (**b**) on tested *Brucella* strains. (**a**): CO-MIC determination by the microdilution method with and without CCCP (10 µg/mL). (**b**): Left two columns, CO inhibition diameter on CAMH plates containing 0.2% DMSO or CCCP (10 µg/mL); right column, growth of strains on CAMH + CCCP (10 µg/mL) without CO as positive control to effect CCCP. The two photographs are not to scale. **CO:** colistin; **CCCP:** carbonyl cyanide 3-chloro phenyl hydrazone; **MIC:** minimal inhibitory concentration; **DMSO:** di-methyl-sulfoxide; **CAMB**: cation adjusted Muller Hinton; **Ctrl:** control bacterial growth.

**Figure 3 ijms-24-02106-f003:**
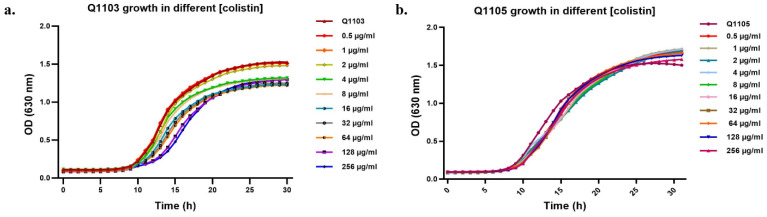
Growth of *B. intermedia* strains treated with different colistin (CO) concentrations. Growth of strain Q1103 was slightly affected by the increasing concentrations of CO (*p* < 0.0388), while growth of the strain Q1105 was not affected (*p* = 0.999).

**Figure 4 ijms-24-02106-f004:**
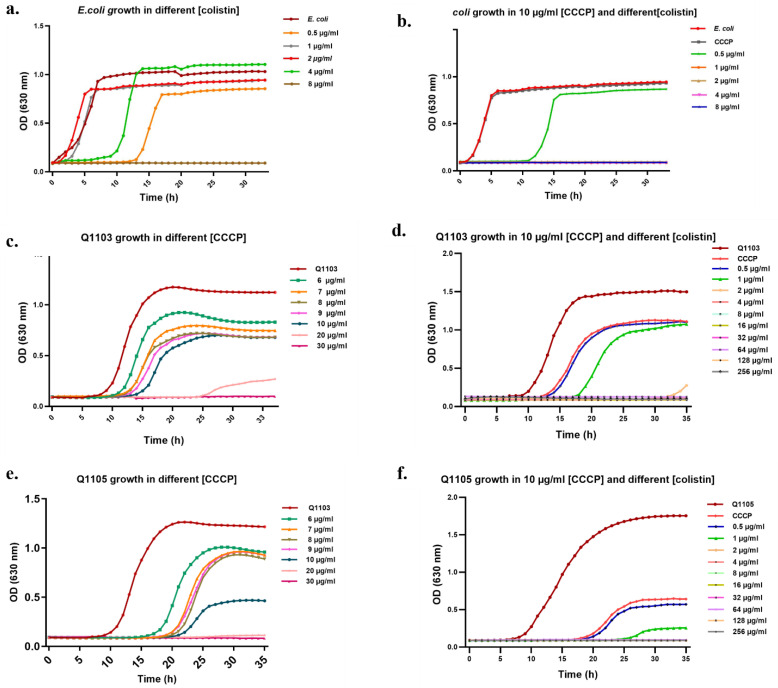
Growth curves of studied strains in different conditions. (**a**) Shows the growth of *E. coli* in different CO concentrations, (**c**) and (**e**) show the growth of *Brucella* strains Q1103 and Q1105 in different CCCP concentrations (**b**) and (**d**,**f**) show the growth of the three strains in different CO concentrations with CCCP (10 µg/mL). Each graph represents the mean of three independent experiments (*p* < 0.0001). **CCCP:** carbonyl cyanide 3-chloro phenyl hydrazone.

**Figure 5 ijms-24-02106-f005:**
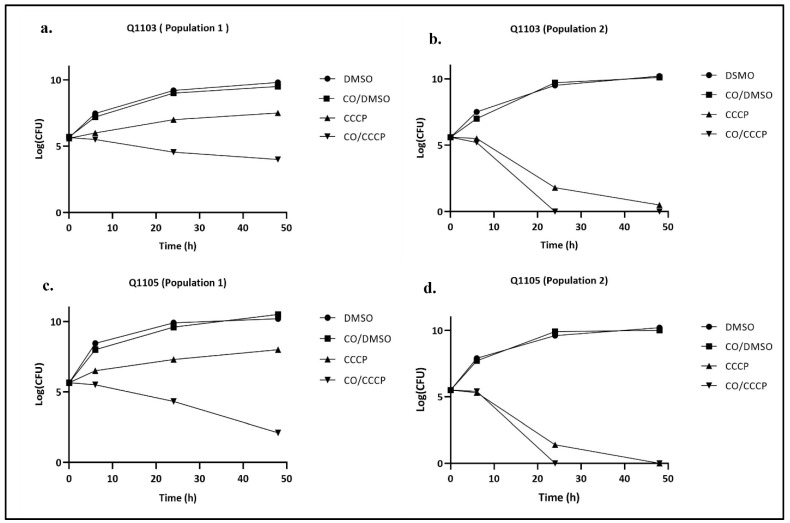
Time-kill curves of the combination of CO (2 µg/mL) and CCCP (10 µg/mL) of populations 1 and 2 of *B. intermedia* strains Q1103 and Q1105. **Population 1:** CCCP-MIC (alone) > 10 µg/mL; **Population 2:** CCCP-MIC (alone) < 10 µg/mL (*p* < 0.05); **CO:** colistin; **CCCP:** carbonyl cyanide 3-chloro phenyl hydrazone; **DMSO:** di-methyl-sulfoxide.

**Figure 6 ijms-24-02106-f006:**
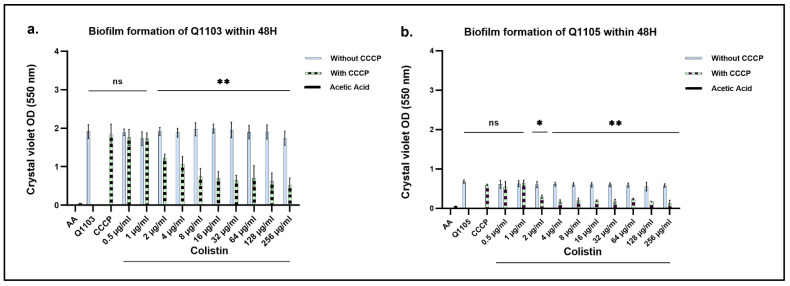
Effect of 10 µg/mL carbonyl cyanide 3-chloro phenyl hydrazone (CCCP) on biofilm formation in *B. intermedia* strains Q1103 and Q1105 after 48 h with and without colistin (CO). ***** and ******: *p* < 0.05; **ns:** non-significant.

**Figure 7 ijms-24-02106-f007:**
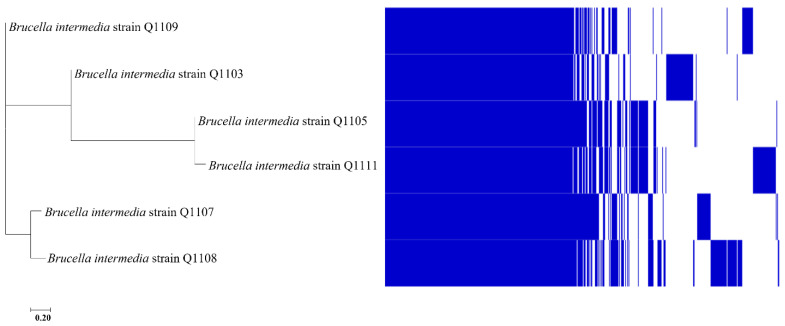
Pangenome analysis of the isolated *B. intermedia* strains carried out using the Roary software. **Left**: core-genome phylogeny, **right**: genes heatmap. The scale represents 20% nucleotide sequence divergence.

**Table 1 ijms-24-02106-t001:** Antibiotic susceptibility of *B. intermedia* strains to CO with and without CCCP by the microdilution method and FIC index interpretation.

	Microdilution Method					
		MIC (µg/mL)			FIC Index	
Strains	CO	CCCP	CO/CCCP	MIC Fold Change	Value	Interpretation
* **E. coli** *	>4–8	>30	<1	4–8	0.1–0.2	Synergism
**Q1103**	>512	<30	<2–4	128–256	0.0	Synergism
**Q1105**	>512	<20	<1	512	0.0	Synergism
**Q1107**	>512	<20	<2	256	0.0	Synergism
**Q1108**	>512	<15	<0.5	512	0.0	Synergism
**Q1109**	>512	<20	<1	512	0.0	Synergism
**Q1111**	>512	<15	<1	512	0.0	Synergism

**CO:** colistin; **CCCP:** carbonyl cyanide 3-chloro phenyl hydrazone; **MIC:** minimal inhibitory concentration; **FIC index:** fractional inhibitory concentration index.

## Data Availability

Genomes of all the six *Brucella intermedia* strains were deposited in GenBank database under the following accession numbers. **Q1103**: JAOXXK000000000; Q1105: JAOXXJ000000000.1.; **Q1107**: JAOXXI000000000.1; Q1108: JAOXXH000000000.1.; **Q1109**: JAOXXG000000000.1; Q1111: JAOXXF000000000.1.

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
