# Peer review of "Carbonyl Cyanide 3-Chloro Phenyl Hydrazone (CCCP) Restores the Colistin Sensitivity in *Brucella intermedia"

_ijms, 2023, doi:10.3390/ijms24032106_

Round 1
Reviewer 1 Report
In recent years, interest in the use of the antibiotic colistin has grown due to the spread of bacteria multiresistant to almost all antibiotics. However, the expansion of the use of colistin leads to an increase in the resistance of bacteria to it. A limited set of drugs for the treatment of resistant bacteria makes it extremely urgent to search for approaches overcoming resistance to colistin. The presented manuscript is devoted to this problem (resistance of Brucella intermedia, a non-fermentative bacterium, which is naturally resistant to polymyxins, including colistin. Several efflux pump inhibitors have been examined with the aim of suppressing resistance toward colistin. A detailed study using various methods was carried out and the restore of susceptibility to colistin in bacteria treated with a CCCP inhibitor was convincingly shown. Moreover, a synergistic effect of the action of the CCCP and CO was found. An inhibitory effect of the CO+CCCP combination was also observed on biofilm formation. The manuscript is well written and is suitable for publication in the IJMS.
I have no significant comments, just a few questions that would be noted in the discussion, if possible.
i) How widespread is the resistance of different bacteria to colistin associated with efflux systems? In the Introduction you mention that the most common is the modification of target (LPS).
ii) Do the authors have any suggestions about a possible mechanism of synergistic influence of the USSR and the SO? Comments and comparison of previously obtained data on synergism for other bacteria would be interesting.
Below there are several minor comments:
Table 1 It is necessary to check the number of significant digits for the FIC Indexes: The threshold value of the criterion are determined as 0.5; 1.0; 2.0 etc The values estimated should have the same number of digits.
Figure 2 It is not clear why the right images are only half shown.
Lines 145-146 Is it possible to say about a significant effect of colistin on bacterial growth if OD is reduced by a maximum of 15%?
Lines 184-185 Figure caption is above the graphs.
Line 307, 379 Remove extra space.
Lines 408-414 4.10. Statistical analysis Check font size.
Line 398 0.5μg/mL; 10μg/mL; Line 407 10μg/mL Add a space.
Author Response
- How widespread is the resistance of different bacteria to colistin associated with efflux systems? In the Introduction you mention that the most common is the modification of target (LPS).
Response: Thank you so much. Several systems/mechanisms involved in the resistance to colistin (two-component systems, antibiotic inactivation, efflux pump systems, mobilized mcr genes, and LPS modification) have been reported. The most common mechanism is currently the LPS modification one. We highlighted “currently” because we believe that: i) emerging new culture and identification techniques like “culturomics” will reveal new species and subspecies which may show further involvement of efflux pump system(s) in the colistin resistance mechanism; ii) scientific reports about the involvement of efflux pumps in colistin resistance are increasing since 2007 and our work is an example where these systems are currently under investigation (mutations and transcriptomics studies) to better decipher their role in colistin resistance. In lines 54 and 55 of the introduction, we added “ as was reported in Acinetobacter sp. and Pseudomonas sp.”
- Do the authors have any suggestions about a possible mechanism of synergistic influence of the USSR and the SO? Comments and comparison of previously obtained data on synergism for other bacteria would be interesting.
Thank you. Sorry but I didn't understand what you mean with USSR and SO to answer.
Below there are several minor comments:
- Table 1. It is necessary to check the number of significant digits for the FIC Indexes: The threshold value of the criterion are determined as 0.5; 1.0; 2.0 etc. The values estimated should have the same number of digits.
Response: Thank you for your comment. The number of digits for the FIC Indexes values was checked in Table 1.
- Figure 2 It is not clear why the right images are only half shown.
Response : Thank you. The images on the right of Figure 2 are only half shown, just because we used the same agar plate for two different strains used as control with CCCP and without Colistin. There is to avoid spending large amounts of CCCP to prepare the agar plates and keep sufficient quantity to continue the manipulations. We modified the Figure 2 in the text and the Figure S1 in the supplement material by cropping the dark half of plates.
- Lines 145-146 Is it possible to say about a significant effect of colistin on bacterial growth if OD is reduced by a maximum of 15%?
Response: Thank you for the comment. Although the statistical test showed a significant effect but you are right, a decrease of 15% will not be significant as growth still considered important in the presence of high concentrations of colistin. The sentence was modified as following in lines 154-155: “ Increasing concentrations of CO did not affect significantly the growth of both strain (P<0.0388; P=0.999) (Figure 3a, 3b)”.The caption of the Figure 3 was also modified as following in line 170: “Figure 3. Growth of B. intermedia strains treated with different colistin (CO) concentrations. Growth of strain Q1103 was slightly affected by the increasing concentrations of CO (P<0.0388), while growth of the strain Q1105 was not affected (P=0.999)”.
- Lines 184-185 Figure caption is above the graphs.
Response: Thank you. The caption was replaced in lines 197 to 201.
- Line 307, 379 Remove extra space.
Response: Thank you. Spaces were removed. Lines 320, 392.
- Lines 408-414 4.10. Statistical analysis Check font size.
Response: Thank you. Font size for Statistical analysis was checked. Line 422
- Line 398 0.5μg/mL; 10μg/mL; Line 407 10μg/mL Add a space.
Response: Thank you. Spaces were added in lines 410, 411 and 419.

Reviewer 2 Report
In this paper, the authors demonstrated for the first time that the natural resistance to colistin (CO) of B. intermedia can be reversed by Carbonyl Cyanide 3-Chloro Phenyl hydrazone (CCCP). However, several studies already demonstrated that the CO resistance was able to be reversed by CCCP in different other Gram-negative bacteria. The precise mechanism of CO-CCCP synergy and antibiotic sensitivity restoration has not been elucidated in the paper and remains unknown. Although authors hint towards efflux pumps - which we would expect knowing the independent mode of action of these drugs -, additional concrete datasets are missing. In my opinion, the very brief pangenome analysis doesn't reconcile the efflux pumps as mode of action and lacks deeper analysis. In summary, this manuscript brings one small novelty but readers might fall short of any clear mode of action.
Major comments:
* It is hard to think about the rationale behind data 4a. measuring the bactericidal activity of colicin on E. coli. E. coli growth is strongly affected with 0.5 ug/mL of colicin but grow better at 2 ug/mL than cells without colicin. Any comments?
* Please explain earlier why Q1103 and Q1105 strains have been chosen.
* The difference between strains Q1103/Q1105 and their different populations is unclear, and the procedure is not explained. Please specify.
* The pangenome analysis is extremely limited and the figure doesn't provide any meaningful information. There are many conserved efflux pumps in bacteria which have been experimentally demonstrated to be involved in the resistance to polymyxins, and sometimes in Brucella itself. Therefore, we would statistically and naturally expect to find some of these genes in the core genome. Other deeper analysis could be performed: do any of these strains contain mutations in these efflux pumps? If so, is the sensitivity different?
Minor comments:
* Add reference S1 in the text for Q1107, Q1109, Q1111 liquid/agar raw data (Fig 2A) in supplemental (Line 94, for example)
* Line 102 : add reference(s)
* Line 104: Q1107 table 1 shows a CCCP-MIC <20, so should be rephrased
* Line 116: add reference 23 so the CO-resistant E. coli strain can be tracked back
* Line 393 : three biological replicates? Please specify.
* Line 396: growth curve is mentioned to be measured at an optical density of 600 nm. However, figures indicate 630 nm. Please rectify.
* Fig. 4. Please use the same color codes between c and e, d and f. It is confusing.
* Fig. 6. I would suggest to reverse the different conditions for each graph for better visualization (starting with CCCP alone and then increasing concentrations of colistin).
Author Response
Major comments:
- It is hard to think about the rationale behind data 4a. measuring the bactericidal activity of colicin on E. coli. coligrowth is strongly affected with 0.5 ug/mL of colicin but grow better at 2 ug/mL than cells without colicin. Any comments
Response 1: Thank you your comment. This is an interesting question. The E. coli strain that we used as positive control contained the mobilized colistin resistance gene (mcr-1) which conferred plasmid-mediated resistance to colistin. In the research work of Baron et al, they found an inhibitory effect of CCCP on mcr plasmid-mediated colistin resistance [1]. Therefore, it was used as a positive control for the positive effect of CCCP. We assume that the mcr-1 gene was not active at 0.5 µg/mL of colistin but rather at a higher concentration, which could explain the growth curve at 0.5 µg/mL of colistin alone (Figure 4a) and the absence of the CCCP effect on the growth curve at the same concentration (Figure 4b).
- Please explain earlier why Q1103 and Q1105 strains have been chosen.
Response 2: Thank you. The following sentence was added earlier in the results, in line 87 as requested: “Strain Q1103 was phenotypically different from other isolated Brucella strains by an important production of extracellular matrix around colonies. The Q1105-Q1107-Q1108-Q1109-Q1111 strains were phenotypically similar. Thus, we selected two strains that were phenotypically different, Q1103 and Q1105 that was chosen randomly among the five remaining ones for the following analyses.”
- The difference between strains Q1103/Q1105 and their different populations is unclear, and the procedure is not explained. Please specify.
Response 3: For each isolated Brucella strain, different MICs were obtained for CCCP by repeating the experiment 10 times with a large difference in MICs. Only the most frequent MICs were noted in Table 1. This led us to think about the heterogeneity of the response to CCCP as a common criterion for all isolated Brucella strains. As a result, we isolated colonies that respond to distinct CCCP concentration (from 5 to 10 µg/mL) and we divided the population in two groups compared to the used concentration in the literature (10 µg/mL): colonies with CCCP MICs <10 µg/mL and those with CCCP MICs >10 µg/mL.
The MICs of the CCCP/colistin combination and the growth curves were evaluated only on the population that initially had a CCCP MIC >10 µg/mL, only for the two selected strains (Q1103 and Q1105) as mentioned in the previous comment. While the bactericidal effect was tested for both populations CCCP MIC >10 µg/mL and CCCP MIC <10 µg/mL to see if there was a bactericidal or inhibitory effect for CCCP alone and in combination with CO in both groups and to clarify if there are differences in these effects.
More explanations were added in paragraph 2.2 as highlighted in the manuscript from lines 102 to 109.
- The pangenome analysis is extremely limited and the figure doesn't provide any meaningful information. There are many conserved efflux pumps in bacteria which have been experimentally demonstrated to be involved in the resistance to polymyxins, and sometimes in Brucella itself. Therefore, we would statistically and naturally expect to find some of these genes in the core genome. Other deeper analysis could be performed: do any of these strains contain mutations in these efflux pumps? If so, is the sensitivity different?
Response 5: Thank you. you are totally right. Our study opened a way to understand the type of resistance mechanism to colistin and provided promising results for further analyses to identify the involved mechanism of resistance. The pangenome analysis restricted the number of candidate genes to be involved in the natural resistance mechanism of B. intermedia and probably excluded the role of Mex and Acr efflux pumps that contributed to CO resistance in Pseudomonas aeruginosa and Acinetobacter baumannii, as they were not located in the core genome of all strains.
The involvement of YejABEF, an ABC transporter, was previously demonstrated in Brucella but I think also that it remains to be confirmed in species belonging to the group previously designated as Ochrobactrum, as there is still confusion in the classification of Ochrobactrum and Brucella [2] [3]
In this work, we limited ourselves to show the effect of CCCP on Brucella strains alone and in combination with CO, to specify the type of effect, whether inhibitory or bactericidal, and to predict the type of resistance mechanism and the common related genes that could be involved. The revelation of the CO resistance mechanism in B. intermedia remains to be elucidated later in a second work by the evaluation of the presence of mutations, as requested, in the concerned genes and by transcriptomic analysis. Deeper analysis to be performed was mentioned in the discussion line 303.
Minor comments:
- Add reference S1 in the text for Q1107, Q1109, Q1111 liquid/agar raw data (Fig 2A) in supplemental (Line 94, for example)
Thank you. The figure references were added in the text in lines 100, 101, 125 and 130
- Line 102: add reference(s)
Thank you. The reference was added in line 109 as requested.
- Line 104: Q1107 table 1 shows a CCCP-MIC <20, so should be rephrased
Thank you for pointing this out. The sentence is rephrased. Lines 111, 112 and 113
- Line 116: add reference 23 so the CO-resistant E. coli strain can be tracked back
Thank you. The reference 23 was added as requested in line 127.
- Line 393: three biological replicates? Please specify.
Thank you for your comment. The followed sentence was added in line 406: “Three biological replicates were performed to confirm reproducibility.”
- Line 396: growth curve is mentioned to be measured at an optical density of 600 nm. However, figures indicate 630 nm. Please rectify.
Thank you for pointing this out. The used optical density is 630 nm. It was rectified in the text line 409.
- 4. Please use the same color codes between c and e, d and f. It is confusing.
Thank you. The color codes were changed as requested in the figure 4c, and e, d and f.
- 6. I would suggest to reverse the different conditions for each graph for better visualization (starting with CCCP alone and then increasing concentrations of colistin).
Thank you for your comment. The different conditions were reversed in the Figure 6, as requested.
References
- Baron, S.A.; Rolain, J.-M. Efflux Pump Inhibitor CCCP to Rescue Colistin Susceptibility in Mcr-1 Plasmid-Mediated Colistin-Resistant Strains and Gram-Negative Bacteria. J. Antimicrob. Chemother. 2018, 73, 1862–1871, doi:10.1093/jac/dky134.
- Hördt, A.; López, M.G.; Meier-Kolthoff, J.P.; Schleuning, M.; Weinhold, L.-M.; Tindall, B.J.; Gronow, S.; Kyrpides, N.C.; Woyke, T.; Göker, M. Analysis of 1,000+ Type-Strain Genomes Substantially Improves Taxonomic Classification of Alphaproteobacteria. Front. Microbiol. 2020, 11, doi:10.3389/fmicb.2020.00468.
- Moreno, E.; Blasco, J.M.; Letesson, J.J.; Gorvel, J.P.; Moriyón, I. Pathogenicity and Its Implications in Taxonomy: The Brucella and Ochrobactrum Case. Pathog. Basel Switz. 2022, 11, 377, doi:10.3390/pathogens11030377.
